



# Concentration-discharge relationships vary among hydrological events, reflecting differences in event characteristics

Julia L. A. Knapp[1], Jana von Freyberg[1,2], Bjørn Studer[1], Leonie Kiewiet[3], and James W. Kirchner[1,2,4]

[1]Department of Environmental Systems Science, ETH Zurich, 8092 Zurich, Switzerland
[2]Swiss Federal Institute for Forest, Snow and Landscape Research WSL, 8903 Birmensdorf, Switzerland
[3]Department of Geography, University of Zürich, 8057 Zurich, Switzerland
[4]Department of Earth and Planetary Science, University of California, Berkeley, CA 94720, USA

*Correspondence to*: Julia L. A. Knapp (julia.knapp@usys.ethz.ch)

**Abstract.** Studying the response of streamwater chemistry to changes in discharge can provide valuable insights into how catchments store and release water and solutes. Previous studies have determined concentration-discharge (cQ) relationships from long-term, low-frequency data of a wide range of solutes. These analyses, however, provide little insight into the coupling of solute concentrations and flow during individual hydrologic events. Event-scale cQ relationships have rarely been investigated across a wide range of solutes and over extended periods of time, and thus little is known about differences and
similarities between event-scale and long-term cQ relationships. Differences between event-scale and long-term cQ behavior may provide useful information about the processes regulating their transport through the landscape.

Here we analyze cQ relationships of 14 different solutes, ranging from major ions to trace metals, as well as electrical conductivity, in the Swiss Erlenbach catchment. From a 2-year time series of sub-hourly solute concentration data we
determined long-term cQ relationships for each solute and compared them to cQ relationships of 30 individual events. The long-term cQ behavior of groundwater-sourced solutes was representative of their cQ behavior during hydrologic events. Other solutes, however, exhibited very different cQ patterns at the event and long-term scale. This was particularly true for trace metals as well as atmospheric and/or biologically active solutes, many of which exhibited highly variable cQ behavior from one event to the next. Most of this inter-event variability in cQ behavior can be explained by factors such as catchment wetness,
season, event size, input concentrations, and event-water contributions. We present an overview of the processes regulating different groups of solutes, depending on their origin in and pathways through the catchment. Our analysis thus provides insight into controls on solute variations at the hydrologic event scale.



## 1 Introduction

The movement of water and solutes through the landscape is inherently coupled. Streamwater chemistry at a catchment outlet differs depending on the flowpaths of water through the catchment and can therefore be considered a "fingerprint" of catchment transport, mixing, and reaction processes. Consequently, studying the response of streamwater chemistry to changes in discharge provides insight into how catchments store and release water and solutes. Changes in solute concentrations as functions of discharge, i.e., concentration-discharge (or cQ) relationships, have commonly been assessed using multi-year time

series of low-frequency (weekly to monthly) streamwater chemistry measurements (Hall, 1970; Godsey et al., 2009; Musolff et al., 2015; Godsey et al., 2019). At this temporal scale, cQ relationships can serve as indicators of hydrologic and biogeochemical processes. Decreasing solute concentrations with increasing flow (often referred to as "dilution behavior", Fig. 1a) have frequently been associated with source limitations, indicating the depletion of finite solute sources in the catchment (Basu et al., 2011) or mixing with more dilute waters. Conversely, patterns of increasing solute concentrations with

discharge are described as "mobilization behavior" resulting from the flushing of solutes, for example from upper soil layers (Fig. 1c). Solute concentrations that vary little across wide ranges of discharge ("chemostatic behavior", Fig. 1b) can result from several mechanisms, including the storage and release of solutes that are not supply limited (Godsey et al., 2009; Basu et al., 2011), the over-printing of source-limitation and mobilization behavior (Cartwright et al., in press), or from large pre-event water contributions to storm runoff (Clow and Mast, 2010).


Because cQ relationships can vary between solutes and catchments, they are frequently employed as descriptors for catchment hydrological behavior. The cQ relationships obtained from long-term, low-frequency data are particularly useful for characterizing the average behavior of a catchment (Clow and Drever, 1996; Godsey et al., 2009; Godsey and Kirchner, 2014). However, these long-term cQ relationships provide limited insight into the coupling of streamwater chemistry and discharge

on shorter time scales, such as during hydrologic events. To better understand hydrologic controls on hydrochemical processes during events requires high-frequency hydrochemical observations, ideally spanning many contrasting storms. High-frequency streamwater sampling is cost- and labor-intensive, and thus most studies are limited to the characterization of individual hydrologic events (the study by Rose et al., 2018, is a rare exception). Recent technological progress in the development of in-situ sensors now allows for several solutes to be monitored at sub-hourly time scales (Rode et al., 2016). Analyses of these

high-frequency measurements have provided substantial insights into biogeochemical processing (Rusjan et al., 2008), solute dynamics (Evans and Davies, 1998; Schwientek et al., 2013), and the temporal evolution of source contributions over the course of individual storm events (Grimaldi et al., 2004). Event-scale studies have also highlighted a general variability in solute responses across storm events that exceeds the variability observed in weekly or monthly grab samples (Bieroza and Heathwaite, 2015; Lloyd et al., 2016). These findings suggest that the controls on solute storage and transport processes on the

event scale may be fundamentally different from those that shape long-term behavior.





Widely available chemical data from in-situ sensors are limited to a handful of solutes, including nitrate, orthophosphate, and dissolved organic matter. While these solutes provide interesting insights into different aspects of catchment processes (Carey et al., 2014; Dupas et al., 2016; Koenig et al., 2017), they are unlikely to characterize all relevant processes regarding solute

mobilization from different parts of the catchment. Furthermore, event-scale cQ behavior is usually not placed into the context of long-term cQ behavior, because streamwater chemistry has rarely been measured at high temporal resolution over long periods. For example, solute concentrations during successive events in wetter years have been shown to be lower than in drier years (Biron et al., 1999), and events in wetter conditions may result in stronger surface water acidification than events following drier conditions (Wellington and Driscoll, 2004). However, because these studies measured solute concentrations

only during individual events, we have no information on the long-term cQ behavior at these sites. Thus it remains challenging to identify controls of cQ behavior on both the event scale and longer time scales.

In this study we used high-frequency measurements of 14 different solutes ranging from major ions to trace metals in a pre-Alpine stream over a period of 2 years. We quantified the long-term cQ relationships using snow-free periods from the entire

2-year data set, and compared them to event-scale cQ relationships of 30 hydrologic events that differed in size, antecedent wetness conditions, and seasonality indicators. Our study aims to explore questions such as 1) how does long-term cQ behavior differ from cQ behavior observed on the event scale? 2) how variable are cQ relationships between individual events? and 3) can inter-event variability in cQ relationships be explained by specific environmental controls?

## 2 Data and methods

### 2.1 Site description

The Erlenbach catchment is a small (0.7 km$^2$), steep catchment spanning an elevation range from 1100 to 1655 m above sea level in the northern Swiss pre-Alps. The underlying geologic formation is Flysch, and the highly layered bedrock consists of limestone, claystone, marl, and shale, as well as conglomerate and calcareous sandstone (Zobrist, 2010). The bedrock is overlain by umbric Gleysols with high silt and clay content in the steeper areas (Schleppi et al., 1998; Xu et al., 2009), and by

mollic Gleysols with a permanently reduced $B_g$ horizon in the flatter areas (Hagedorn et al., 2000). The catchment landscape is characterized by interchanging slopes and plateaus, and the groundwater table is generally shallower under the plateaus than under the slopes (Rinderer et al., 2014). The soil and bedrock permeabilities are relatively low, resulting in highly saturated soils, particularly on the plateaus. In total, 53 % of the catchment is forested, and dry and wet meadows cover roughly 14 and 33 % of the catchment area, respectively (van Meerveld et al., 2018). Coniferous forests cover the majority of the slopes,

whereas meadows and partially forested areas can be found on the plateaus. Agricultural influence is limited to summer-season cattle grazing in the upper part of the catchment. Average annual precipitation in the Erlenbach catchment is 2300 mm, of which up to 40 % falls as snow in the winter months (Stähli and Gustafsson, 2006), and about 20 % of incoming precipitation



leaves the catchment as evapotranspiration (van Meerveld et al., 2018). In the years 2017 and 2018, stream discharge at the catchment outlet ranged from 0.2 to 2240 L s$^{-1}$, with an average value of 37.7 L s$^{-1}$.


## 2.2 Dataset

Streamwater and precipitation chemistry were analyzed semi-continuously by an automated field laboratory located at the Erlenbach catchment outlet (von Freyberg et al., 2017; von Freyberg et al., 2018). Streamwater was pumped continuously from the stream to the field laboratory, and precipitation was collected with an open 45 cm diameter funnel. During periods without

rain, the field laboratory received only streamwater for analysis. The field laboratory analyzed precipitation whenever more than 50 ml of precipitation accumulated in the rain sampler (corresponding to roughly 0.3 mm of precipitation), and the previously analyzed sample was streamwater. A new sampling and analysis cycle was initiated every 30 minutes; thus, during rainfall, streamwater and precipitation samples were each analyzed hourly. A drift correction standard was analyzed instead of streamwater every 4 hours (every 6 hours after 12$^{th}$ March 2018).


Before analysis, each precipitation or streamwater sample passed through a 0.2 μm tangential filter. One aliquot of each sample was automatically directed to an ion chromatograph (940 Professional IC Vario, Metrohm AG, Herisau, Switzerland, hereafter referred to as "IC") for the analysis of major anions and cations (calcium, magnesium, sodium, potassium, chloride, nitrate, and sulfate). Another aliquot was injected into a continuous water sampler module (CWS, Picarro Inc., Santa Clara, CA, USA)

coupled to a cavity ring-down spectroscope (CRDS, Picarro Inc., Santa Clara, CA, USA) for water isotopes analysis (deuterium and oxygen-18). Further details on the sampling and analysis of isotopes and major ions in the field laboratory is described by von Freyberg et al. (2017) and von Freyberg et al. (2018).

In addition to the on-site analysis of major ions and stable water isotopes, aliquots of filtered rain water (every sample) and

streamwater (one sample per hour) were automatically collected into vials. Each vial contained 1 ml of ultrapure HNO$_3$ to stabilize the water sample. These acidified samples were collected approximately once per week and brought to the laboratory at ETH Zurich for subsequent analysis of a wide range of trace elements using inductively coupled plasma mass spectrometry (Agilent 7900 ICP-MS, Agilent Technologies, Santa Clara, CA, USA, hereafter referred to as "ICP-MS"; more detail on the laboratory protocol can be found in the SI). Of the measured elements, we selected boron, strontium, barium, iron, manganese,

copper, and chromium for further analysis in this study. The other elements were not included here because their concentrations were mostly below the analytical detection limits. Outlier removal of isotope, IC, and ICP-MS measurements was based on visual inspection.



Streamwater electrical conductivity (EC) was measured at 5-minute resolution (s::can condu::lyser, scan Messtechnik GmbH,
Vienna, Austria). For the purpose of this study, the EC data were aggregated to 10-minute intervals to match the measurement
frequency of river discharge (see below).

River discharge was measured at a concrete flume installed at the Erlenbach catchment outlet (Hegg et al., 2006). Precipitation
rates were recorded with a heated tipping-bucket rain gauge (Joss-Tognini 15183, Lambrecht meteo GmbH, Göttingen,
Germany) at a meteorological station located at 1216 m above sea level in the Erlenbach catchment. At the meteorological
station, air temperature was measured with a ventilated thermometer (VT3, Meteolabor AG, Wetzikon, Switzerland), and
groundwater level fluctuations were recorded in a fully screened well (these readings were relative rather than absolute,
because they have not been calibrated against manual water level measurements). Discharge, precipitation, air temperature,
and groundwater levels were recorded at 10-minute resolution. These data were aggregated to 30-minute or 1-hour intervals
to match the frequency of the solute data, except for the analysis of the EC data, for which the 10-minute resolution was used.

### 2.3 Long-term dataset and event identification

The Erlenbach stream generally shows a fast and flashy response to rainfall events, resulting in very short durations of the
rising limb of the storm hydrograph, in particular during short and intense events. Consequently, few samples were collected
during the rising limb of each storm in spite of the high sampling frequency of the field laboratory. The falling limb of the
streamflow hydrograph, on the other hand, was generally well captured. For this reason, we excluded all samples collected
during periods of increasing discharge from our analysis (in Sect. 3.3 below, we show that excluding these periods has
negligible effects on the long-term cQ behavior). The relationship between solute concentrations and streamflow during
hydrograph recession is informative in particular on outflow processes from shallow and deeper groundwater, as well as
riparian water (Inamdar et al., 2006).

We excluded the winter months from the dataset to avoid ambiguities arising from rain-on-snow and snowfall events in our
analysis, because solute responses to these events may differ from those during rain events in snow-free periods. Consequently,
only data points between May 1st and November 15th in both 2017 and 2018 were analyzed (hereafter referred to as the "snow-
free periods"). During these snow-free periods, we identified individual events based on the following criteria: (1) events had
to have a substantial increase in discharge, i.e. peak discharge at least 20 L s$^{-1}$ above pre-event baseflow; (2) this discharge
increase had to be triggered by rainfall; (3) the hydrograph had to recede by at least 75% of the absolute increase in discharge
before the event was cut off (e.g. due to a subsequent rain event or sampler failure); and finally, (4) only events were considered
for which IC (i.e., major ions) measurements were available.






For each event, we analyzed the solute concentration and discharge data during the hydrograph recession, starting at the main discharge peak of an event until 95% hydrograph recession to baseflow, or until the event was cut off by the start of a subsequent event (if discharge had receded by at least 75%). We estimated cQ relationships for solute/event combinations for which at least 5 data points were available on the recession limb.

## 2.4 Assessment of source-area concentrations and input-output budgets

To identify likely streamwater sources, we quantified average solute concentrations of different compartments in the Erlenbach catchment: median solute concentrations in streamwater and precipitation and their lower and upper quartiles were calculated from the snow-free periods in the time series. Solute concentrations in groundwater were obtained from sampling two pumping wells in the upper part of the catchment at three different instances between 2016 and 2017. Although these groundwater chemistry data may not provide a comprehensive picture of Erlenbach groundwater, they can nonetheless indicate which solutes are likely to be dominant therein. Concentrations in soil water of the neighboring Studibach catchment were obtained from sampling eighteen suction lysimeters during eight baseflow snapshot campaigns in the snow-free season in 2016 and 2017 (detailed description of the campaigns are given in Kiewiet et al., 2019). We aggregated the data from six sites (three forested, three non-forested), spread over three elevations in the catchment (1361, 1502 and 1611 m above sea level), at which suction lysimeters were installed at 15, 30 and 50 cm depth. The lysimeters were emptied and set to a tension of 50 mbar the day prior to sampling.

We characterized the source-sink behavior of the Erlenbach catchment for individual solutes using a dimensionless solute flux index calculated from the high-frequency data of the snow-free periods. The solute flux index $F$ relates solute fluxes in precipitation inputs $I$ to those in streamwater outputs $O$:

$$F = \frac{O - I}{\max(O, I)} = \frac{Q_{tot}\langle c_{Q_i}\rangle - P_{tot}\langle c_{P_i}\rangle}{\max(Q_{tot}\langle c_{Q_i}\rangle, P_{tot}\langle c_{P_i}\rangle)} \tag{1}$$

Where $Q_{tot}$ and $P_{tot}$ are the total streamflow and precipitation water fluxes, $c_{Q_i}$ and $c_{P_i}$ are the solute concentrations in streamflow and precipitation for all sampling times $i$, and the angled brackets indicate volume-weighted averages. The index is positive if the streamwater solute flux is larger than the precipitation solute flux, and quantifies the fraction of the output flux $O$ generated within the catchment. Conversely, if the index is negative, it quantifies the fraction of the input flux $I$ retained within the catchment. If the input and output fluxes are exactly balanced, the flux index will be zero. Importantly, the solute flux indices as calculated here provide no information on long-term fluxes, but relate the output and input fluxes during the snow-free periods between May and November from which the hydrologic events were extracted.





## 2.5 Quantification of concentration-discharge relationships

We estimated both the long-term cQ behavior (i.e., using all recession data from the snow-free periods in 2017 and 2018), and the individual event-scale cQ behavior by fitting power-law relationships between concentration $c$ and discharge $Q$ to the data
(Clow and Drever, 1996; Musolff et al., 2015):

$$c = aQ^b \qquad (2)$$

This power-law relationship is identical to a linear relationship in double-logarithmic space:


$$\log_{10}(c) = \log_{10}(a) + b\log_{10}(Q) \qquad (3)$$

Where $\log_{10}(a)$ and $b$ are the intercept and slope of the cQ relationship, respectively. cQ slopes and intercepts of the entire dataset will be referred to as "long-term" cQ slopes and intercepts, whereas "event-scale" will refer to cQ relationships of
individual hydrologic events. For the purpose of this study, we normalized discharge by the average discharge of the time series ($\log_{10}(Q/Q_{mean})$) to obtain long-term intercepts that reflect the expected concentration at the mean discharge, rather than the arbitrary value of $Q$=1 ($\log_{10}Q = 0$). Centering the x-axis in this way also has the benefit of making the slope and intercept estimates more independent from one another.

We calculated the relative standard errors of the event-scale slopes and intercepts, in order to exclude events for which the cQ relationships could not be well constrained. cQ relationships were excluded from our analysis if the relative standard error of either the cQ slope *or* intercept for any event and solute exceeded 50 %, or if both the relative standard error of the cQ slope *and* intercept exceeded 25 %.

Given that our samples represent chemistry of hydrograph recession, the meaning of the obtained cQ slopes can be interpreted as follows: a cQ slope of 1 (or -1) is obtained if changes in solute concentrations are proportional (or inversely proportional) to changes in discharge during recession. Consequently, a cQ slope between -1 and 1 indicates less-than-proportional changes in solute concentrations, and a cQ slope close to zero indicates solute concentrations that change relatively little, or that vary independently of discharge during recession. Conversely, cQ slopes greater than 1 indicate that solute concentrations decrease
more-than-proportionally to discharge during recession. Examples of the relationships between cQ slopes, solute recessions, and hydrograph recessions are illustrated in Fig. 1.





**2.6 Possible environmental controls of inter-event variability in cQ relationships**

The 30 hydrologic events span wide ranges of storm durations, intensities, antecedent wetness conditions, and other potential
controls, thus facilitating an investigation into how these environmental controls may influence the slopes and intercepts of
the event-based cQ relationships. To this end, we quantified 15 different parameters for each event from the following five
categories: (1) temperature and proximity to midyear as seasonality indicators; (2) relative input concentrations, which quantify
the ratio between the volume-weighted average precipitation concentration during the event to the streamwater solute
concentrations during pre-event baseflow; (3) groundwater levels, baseflow discharges, and antecedent precipitation as
indicators of antecedent wetness conditions; (4) several measures of event magnitude and intensity; and (5) event and pre-
event water contributions determined from isotope hydrograph separation (following the approach presented by von Freyberg
et al., 2018). Table 1 presents an overview of these 15 environmental controls, as well as their ranges in our dataset.

As a seasonality indicator, the proximity to midyear is calculated as a summer-winter index:

$$SW = \begin{cases} \dfrac{doy}{182.5}, & doy < 183 \\ \dfrac{365 - doy}{182.5}, & doy \geq 183 \end{cases} \tag{4}$$

Where $doy$ is the day of year. This summer-winter index approaches 0 at the beginning and end of each calendar year, and
approaches 1 at the beginning of July.

We used weighted rank correlation coefficients to quantify the dependence of event-scale cQ slopes and intercepts on the 15
environmental controls. The weights were the inverses of the standard errors of the individual cQ slopes and intercepts, to
prevent highly uncertain points from substantially influencing the results. The statistical significances of these correlation
coefficients (their *p*-values) quantify the probability of obtaining an equal or greater correlation if the null hypothesis were
valid (i.e., if there were actually no relationship between the slope or intercept and the respective control).

**3 Results and discussion**

**3.1 Dataset**

We extracted 30 events from the time series of 2017 and 2018 that fulfilled the criteria outlined in Sect. 2.3. While IC
measurements were available for all 30 events, fewer events had sufficient ICP-MS measurements (i.e., for boron, barium,
iron, manganese, chromium, strontium, and copper), because concentrations of these elements were often very low. Between
24 and 30 events were available for major ions, but only 11 events were analyzable for copper and 17 for manganese. All other
solutes yielded analyzable data for more than 20 events.



We quantified most environmental controls for all 30 events. However, meaningful event-water contributions could only be calculated for 22 events from stable water isotope measurements. Ratios of precipitation concentration to the pre-event baseflow solute concentration were not available for all events and all solutes due to sporadic problems with the rain collector.
Also, these ratios were not assessed for EC, because EC was not measured in precipitation. Table 1 provides an overview of the range of environmental controls covered by the selected events. Some small events lasted only a few hours, whereas other events were extended, multi-day storms, and antecedent wetness conditions ranged from relatively dry to very wet. Although all events took place between May and November, they spanned a wide range of air temperatures and weather conditions.

### 255 3.2 Characterization of solute contributions from different source areas

Streamwater concentrations at Erlenbach were dominated by calcium, sulfate, magnesium, and sodium, with median concentrations between 2 and 48 mg L$^{-1}$ (Table 2). Conversely, median streamwater concentrations of manganese and copper were low at around 1 μg L$^{-1}$, and concentrations of chromium were even lower. All other solutes (strontium, barium, boron, chloride, potassium, nitrate, and iron) were observed at intermediate concentrations in the Erlenbach streamwater.


We can group different solutes based on their most important sources in the Erlenbach catchment as determined from their concentration ranges in groundwater, streamwater, and precipitation, as well as output-input flux indices (Table 2). The concentrations of calcium, strontium, barium, and boron in the groundwater samples were similar to, or higher than, those in streamwater. Concentrations of magnesium, sodium, and sulfate were higher in streamwater than in the groundwater samples,
suggesting that other, unsampled, groundwaters with higher concentrations also contribute to streamflow. Groundwater concentrations of weathering products are likely to be highly heterogeneous due to spatially variable contributions from the geochemically complex Flysch bedrock (Fischer et al., 2015; Kiewiet et al., 2019). Precipitation concentrations of these weathering products are low, and flux indices close to 1, implying that they are primarily derived from within-catchment processes (Table 2). In this paper, we will refer to these solutes as groundwater-sourced.


The output-input flux index of chloride at Erlenbach was 0.33 for the snow-free periods, but was -0.13 if calculated from daily data over the two full years (not shown here), suggesting that precipitation is the most important source of chloride in the Erlenbach catchment on an annual basis (Zobrist, 2010). Chloride is likely to be relatively unreactive in the Erlenbach catchment.


Nitrate concentrations in Erlenbach streamwater can also be mainly attributed to atmospheric inputs as well as manure inputs from grazing. The output-input flux index of nitrate was -0.52 (even without taking manure inputs into account), indicating that nitrate was either taken up into vegetation or possibly volatilized as ammonia (i.e., the catchment acted as solute sink). Potassium is supplied to the stream through weathering of the bedrock, with some atmospheric inputs, which contribute more





to streamflow fluxes of potassium than those of calcium or magnesium. Potassium is also known to be retained in the soils and cycled internally in forest stands (Hornung et al., 1986; Likens et al., 1994).

Iron was the most abundant trace metal in the Erlenbach streamwater, with median concentrations of around 5 µg L$^{-1}$. Streamwater iron concentrations were substantially higher than the concentrations measured in groundwater or precipitation,
and very high in the soil water of the neighboring and geologically similar Studibach catchment, pointing toward the soil layer as predominant source of streamwater iron (Table 2). Groundwater concentrations of manganese at Erlenbach were broadly similar to those measured in streamwater, but soil water concentrations (and also groundwater concentrations; not shown here) in the Studibach catchment were substantially higher. This indicates that manganese may be sourced both from groundwater and soil water. The storage of iron and manganese in and release from soil layers are likely controlled by the complexation of
these trace metals with organic material (Bloomfield, 1953; Harter and Naidu, 1995), and the strong redox-sensitivity of these elements (Drever, 1988; Basu et al., 2010; Koger et al., 2018). The concentrations of the trace metals chromium and copper were low in all analyzed compartments, and a distinct source cannot decisively be identified. In the neighboring Studibach catchment, Kiewiet et al. (2019) observed the highest concentrations of chromium at predominantly dry sites, and highest concentrations of iron and manganese at predominantly wet sites. These observations suggest that concentrations of trace
metals are probably not homogeneously distributed in the Erlenbach catchment.

### 3.3 High inter-event variability that differs from the long-term behavior

We calculated long-term cQ relationships based on the snow-free recession periods of the 2-year time series, and event-scale cQ behavior from the recessions of the 30 extracted events. Long-term cQ relationships were relatively narrow and well-
defined for most solutes (Fig. 2), resulting in low uncertainties of long-term slopes and intercepts (Table S1). Furthermore, long-term slopes and intercepts calculated for the whole time series were broadly similar to those determined from only hydrograph recession periods (Table S1), largely because recessions comprised nearly all of the long-term data. Thus our analysis can straightforwardly compare cQ behavior during individual recessions with the long-term cQ behavior across two years of recessions.


Figure 2 shows that major ions and EC exhibited dilution behavior (slopes < 0; lower concentrations at higher discharges) across the long-term record, but with different degrees of variability. The data clouds in the cQ space were more scattered for e.g. chloride, potassium, and nitrate compared to those of e.g. calcium, magnesium, and sodium. Iron, manganese, and chromium showed a mobilization behavior (long-term slope > 0; higher concentrations at higher discharges), while the long-
term cQ relationship of copper indicated chemostasis (slopes ≈ 0).



Event-scale cQ relationships at Erlenbach were much more variable than the long-term behavior. A comparison between different events revealed substantial inter-event variability in both slopes and intercepts for a number of solutes (colored lines in Fig. 2 and blue circles in Fig. 3). The extent of this variability differed dramatically among the solutes. Individual events

followed the long-term trend for calcium, magnesium, sodium, strontium, and EC, for example, whereas they deviated substantially from the long-term trend for potassium, chloride, and nitrate.

For most solutes, the averages of all event-scale intercepts (light-blue diamonds in Fig. 3a) were similar to the long-term intercepts (red bars in Fig. 3a), because discharge was normalized by its mean value (see Sect. 2.5). The inter-event variability

in intercept values was particularly high for chloride, nitrate, and some of the trace metals, possibly suggesting changes in sources, flowpaths or reaction rates of these constituents between events.

Conversely, the slopes of the event-scale cQ relationships differed substantially from the long-term behavior for some of the elements (Figs. 2 and 3). This was true in particular for three solutes with significant atmospheric inputs (chloride, potassium,

and nitrate), which exhibited dilution behavior on the long-term (long-term slope < 0), while their cQ slopes of individual events varied between dilution (event slope < 0) and mobilization (event slope > 0).

None of the solutes exhibited stronger dilution behavior, on average, during events (blue diamonds in Fig. 3b) than over the long term (red bars in Fig. 3b). In the case of potassium, chloride, nitrate, and manganese, most individual events (blue circles

in Fig. 3b) exhibited substantially weaker dilution, or greater mobilization, than the long-term cQ relationship (red bars in Fig. 3b). Considered together, these results indicate that dilution processes were equally important, or more important, over the long term than during individual events.

Solutes with similar sources and similar chemical properties tend to exhibit similar long-term cQ behavior and similar degrees

of inter-event variability in slopes and intercepts. For example, calcium, magnesium, sodium, strontium, barium, boron, sulfate, and EC all exhibited well-defined long-term dilution behavior and little variability in both cQ slopes and intercepts at the event scale. All solutes exhibiting this behavior were previously identified as groundwater-sourced in the Erlenbach catchment (Sect. 3.2). Average event slopes of these solutes were approximated well by the long-term behavior (Fig. 3b), resulting in event patterns that fan out around the high-discharge section of the long-term cQ trend (Fig. 2). This similarity between long-term

and event-scale behavior indicates that similar mechanisms controlled solute and water mobilization both during individual events and over the long term for these solutes.

Conversely, potassium, chloride, and nitrate have substantial atmospheric inputs at Erlenbach. These three solutes were characterized by long-term dilution behavior, and high variability in event slopes ranging from negative to positive values,

along with some inter-event variability in intercept values. These solutes exhibit event cQ patterns that fan out and stack up



around the long-term relationship. This large inter-event variability in cQ slopes may plausibly arise from varying degrees of atmospheric deposition, resulting in temporally variable concentrations of chloride, potassium, and nitrate in shallow soil layers. A large variability in soil water concentrations was indeed observed in the neighboring Studibach catchment (Table 2). In the case of nitrate and potassium, the inter-event variability may also be affected by biological uptake and reaction processes.

In summary, chloride, nitrate, and potassium showed a much more diverse behavior on the event-scale and also – at least to some extent – on longer timescales than the purely groundwater-sourced solutes.

The trace metals iron, manganese, and chromium were mobilized during events, as reflected in positive event and long-term cQ slopes. Interestingly, this mobilization behavior was more pronounced (i.e. slopes were steeper) on the event scale than on

longer timescales. In contrast to the other metals, copper concentrations in streamwater suggested chemostatic behavior over long timescales, whereas some mobilization behavior was evident on the event scale (Fig. 2). These findings suggest that the trace metals were mainly sourced from the soil layers, and the observed cQ behavior of manganese and iron confirms common assumptions about them being co-located in the soil layers where they are bound to organic material.

The grouping of solutes based on cQ behavior aligns well with the previously identified source areas in the Erlenbach catchment (Sect. 3.2). These alignments were most obvious for the purely groundwater-sourced solutes, because they could be clearly attributed to one main source. For most other solutes, the combination of different sources and the co-occurrence of reaction and mixing processes resulted in less clear patterns. A scatterplot of event slopes against event-intercept values (Fig. 4) supports the grouping of solutes and illustrates the different degrees of inter-event variability. The low uncertainty of the

calculated slopes and intercepts furthermore confirms that the observed variability among events is real-world behavior rather than noise.

Other likely modulators of the variability observed for different solutes are their ionic form and their possible occurrence as nanoparticulates. Cations are known to undergo exchange buffering through electrostatic binding to negatively charged sites

in soils (Helling et al., 1964; Rhoades, 1982). This likely resulted in less variable cQ behavior on the event scale for the cationic solutes calcium, magnesium, sodium, strontium, and barium compared to the anionic solutes sulfate and boron. These anions are less buffered by ion exchange and showed a far more variable cQ behavior. Two other anions, chloride and nitrate, also exhibited highly variable cQ behavior at the event scale. It consequently seems likely that exchange buffering processes modulate the degree of observed inter-event variability. Nevertheless, the importance of cation exchange buffering on the

variability of solute behavior cannot be well constrained with our data set, because most analyzed cations are groundwater-sourced species while two of the three solutes with substantial atmospheric inputs, chloride and nitrate, are anions. The only cation with important atmospheric input (potassium) is biologically very active, potentially obscuring the effects of cation exchange buffering. With respect to the trace metals, a large part of their total concentration can occur as natural



nanoparticulates (Gottselig et al., 2017), which may undergo different transport and adsorption processes compared to their
dissolved forms. This likely had a strong effect on their cQ behavior at Erlenbach.

## 3.4 Environmental controls of the observed inter-event variability in cQ relationships

We used weighted rank correlation coefficients to assess how variations in cQ slopes and intercepts from event to event were
related to seasonality indicators, relative input concentrations, antecedent wetness conditions, event characteristics, and event-
water contributions. A heatmap (Fig. 5) illustrates how event-scale cQ slopes and intercepts depended on these different
environmental controls. Individual examples of these relationships are shown in Fig. S1 for different solutes and drivers.

The event-scale cQ intercepts for all solutes, regardless of their dominant source, generally responded similarly to each control
(with the exception of chloride; Fig. 5a). Seasonality indicators and antecedent wetness conditions were the most important
environmental controls on cQ intercepts for all solutes, with drier and warmer conditions resulting in higher concentrations in
streamwater. By contrast, although the effects of each control on event-scale cQ slopes were broadly similar within groups of
solutes that shared the same dominant source (precipitation, soil water, and groundwater), they often differed between these
groups (Fig. 5b). In the following, we therefore discuss the behavior of the event-scale cQ slopes of these groups of solutes
together, rather than discussing individual solutes.


We found that the event-scale cQ slopes of groundwater-sourced solutes were positively correlated with seasonality indicators,
with weaker solute dilution (i.e., less negative cQ slopes) during warmer conditions. Antecedent wetness conditions and event
characteristics were other important factors controlling the event-scale cQ behavior of most groundwater-sourced solutes. We
found stronger dilution (i.e., more negative cQ slopes) during larger events associated with wetter antecedent conditions. cQ
slopes were also more negative when event-water contributions were larger, consistent with stronger dilution of groundwater-
sourced solutes by larger volumes of recent rainfall (i.e., event water). Most of the observed relationships were less clear for
boron (which is primarily sourced from groundwater, but does not occur as a cation, but instead as either undissociated boric
acid or as the borate anion, depending on pH), supporting the hypothesis that differences in ionic charge modulate event-scale
cQ behavior. By contrast, however, sulfate (an anion) showed very similar dependencies as the cations in groundwater,
contradicting this hypothesis. Streamwater electrical conductivity (EC) at Erlenbach is dominated by the two most abundant
solutes, calcium and magnesium (along with their counter-ion, bicarbonate), which are groundwater-sourced. Consequently,
the event-scale cQ behavior of EC depends on similar factors as that of the groundwater-sourced solutes.

For the solutes with significant atmospheric inputs (chloride, potassium, and nitrate), the effects of possible environmental
controls on cQ slopes were less clear, but still indicated similar behavior among the solutes of this group. We found that the
event-scale cQ slopes of chloride and nitrate were influenced by antecedent wetness conditions, with dilution behavior during



wetter conditions and mobilization behavior during drier conditions (Fig. 5b). This can potentially be explained by stronger evapo-concentration of these solutes in the soil under drier conditions. We also observed a tendency toward more positive slopes with higher relative input concentrations for all three solutes, and during events in which the event-water contribution was larger. The observed dependencies of nitrate and potassium differed somewhat from those of chloride, likely due to the overprinting with reaction processes acting on these solutes, and due to overlapping patterns from atmospheric and groundwater sources in the case of potassium.

Among the trace metals, iron often exhibited contrasting behavior compared to the others (Fig. 5b). While manganese, chromium, and copper showed a tendency toward stronger mobilization under colder and wetter conditions, iron tended to be mobilized more during warmer and drier conditions. Given that both iron and manganese are found predominantly in wetter soils (in the neighboring Studibach catchment; Kiewiet et al., 2019), we would have expected these two metals to depend on similar drivers. However, Kiewiet et al. (2019) also observed higher concentrations of manganese in riparian areas compared to iron. We would expect these riparian zones to be activated first during a rain event, whereas soils farther from the stream would likely only start to contribute to streamflow later. This sequence of contributions from source areas dominated by different trace metals could potentially explain why the cQ slopes of the trace metals varied differently with seasonality, event characteristics, and event-water contributions. The high concentrations of manganese in riparian areas may also explain the very high cQ slopes observed for this element. As the relative contribution from riparian areas decreases later in the recession, manganese concentrations in streamwater drop rapidly, decreasing more-than-proportionally compared to discharge.

## 4 A classification scheme for cQ relationships in the Erlenbach catchment

Previous studies have shown that hydrological controls on streamwater chemistry are mostly invariant on time scales of weeks to months; for example, Gwenzi et al. (2017) found similar cQ behavior for weekly to monthly sampling frequencies, and Godsey et al. (2019) found similar cQ behavior in weekly/monthly grab samples and in year-to-year variations in annual average concentrations. Other studies have also shown that long-term average cQ behavior is relatively independent of the sampling frequency (e.g., Bieroza et al., 2018). Our analysis of event-scale cQ patterns, as defined by high-frequency sampling within and between individual events, provides a new perspective on cQ relationships. Our results indicate that solute responses to discharge variations can be fundamentally different during individual events compared to longer time scales. These differences in cQ behavior reveal the effects of the dominant sources, reaction processes, and ionic forms of the different solutes.

Figure 6 provides an overview of the different cQ patterns observed at Erlenbach, and the role of various environmental controls in shaping these patterns during events and over the long term. For groundwater-sourced solutes such as calcium, magnesium, and sodium, we found that long-term cQ relationships were relatively good approximations of event-scale cQ





patterns. These solutes exhibited little inter-event variability, and their cQ relationships on both the event scale and longer time
scales were dominated by dilution. The inter-event variability of cQ slopes of groundwater-sourced solutes was mainly
controlled by season, event size, and event water contributions.

Atmospherically derived solutes also exhibited long-term dilution behavior, but their event-scale behavior ranged from dilution
to mobilization, with event-scale cQ patterns stacking up and fanning out around the long-term cQ relationship (Figs. 2 and
6). Their event-scale cQ slopes were usually more positive than their long-term cQ slopes, indicating a stronger importance of
chemostatic and mobilization mechanisms on the event scale. The controls on event-scale cQ slopes were less clear for nitrate
and potassium than for chloride, likely due to reaction processes (in the case of potassium and nitrate) or overprinting of
contributions from different sources (in the case of potassium). Nevertheless, we observed stronger mobilization following
drier antecedent conditions and during events with larger event-water contributions, suggesting that evapo-concentration of
atmospherically deposited solutes plays an important role.

Trace metals showed mixed behavior in our analysis, likely due to different patterns of distribution in soils and groundwaters,
and possibly also due to their presence as nanoparticulates (Fig. 6). In our study, the different trace metals often responded
differently to environmental controls, possibly reflecting differences in their relative abundance in the riparian zone, in their
complexation mechanisms, in their redox sensitivities, and in their biological cycling (Herndon et al., 2015; Koger et al., 2018).

## 5 Summary and conclusions

Our analysis of 30 events extracted from a 2-year time series of sub-hourly streamwater solute measurements demonstrated
that concentration-discharge (cQ) relationships at the hydrologic event scale can differ substantially from those over the long
term. In addition, cQ relationships varied greatly from event to event for some solutes (e.g., potassium, chloride, and nitrate),
but varied much less for others (e.g., calcium, magnesium, sodium, and EC). The variability in cQ relationships among different
hydrologic events (and solutes) could be linked to a range of environmental controls.

Our analysis would not have been possible if we had analyzed only a few solutes or collected data only during a handful of
hydrologic events, as is common practice in catchment hydrochemistry studies (Rode et al., 2016). To understand the complex
mechanisms governing solute storage and release from different parts of the catchment, it is necessary to analyze many
different solutes, and to sample multiple hydrologic events at high enough frequency to capture event behavior. However, we
are aware that sampling and analysis systems like the one that we employed at Erlenbach are resource-intensive, and thus
difficult to deploy in many field situations. Our analysis suggests, however, that a viable alternative may be to analyze one
solute (or proxy thereof) from every major store and streamwater source in the catchment. For instance, EC in Erlenbach
streamwater exhibited very similar cQ relationships as calcium, magnesium, and sodium, making it a suitable tracer for

groundwater-sourced solutes (in other catchments, EC may be a better proxy for solutes from other compartments; Benettin and Van Breukelen, 2017). Automated sensors for nitrate and phosphorous are available and can provide high-frequency

information on biogeochemically active solutes. The analysis of trace metals is not available through automated sensors, but in many cases DOC can be a suitable proxy for iron and some other trace metals (Nierop et al., 2002; Grybos et al., 2007). All in all, with sufficient knowledge about the possible sources that contribute to streamflow in a catchment, much simpler measurement systems may still provide meaningful insight into solute storage and release processes.

The grouping of solutes based on their dominant source, cQ behavior, ionic character, and dependence on environmental controls is illustrated in a schematic overview in Fig. 6. This overview highlights similarities and differences among solutes, summarizes their expected range of cQ behavior on the event scale, and indicates their sensitivities to environmental controls. Analyses of event-scale cQ patterns may help in identifying the vulnerability of different catchment compartments to changes in land use and climate, and may benefit monitoring and management strategies. For example, if the climate warms and

summers become drier in this area of Switzerland, our data suggest that the event slopes of chloride, nitrate, and potassium will become more positive, leading to enhanced flushing of these solutes during hydrologic events. Evaluating the generality of the results presented here will require further studies in catchments of contrasting climate, geology, and land use.

**Data availability**

The data that support the findings of this study are available from the corresponding author upon reasonable request.

**Author contribution**

JLAK and JWK conceptualized the study. JLAK, JF, and BS collected and analyzed the solute data and LK the soil moisture

data from Studibach, JLAK analyzed the data set, and JLAK prepared the manuscript with contributions from all co-authors.

**Competing interests**

The authors declare that they have no competing interests.

**Acknowledgements**

The authors thank the Swiss Federal Institute for Forest, Snow and Landscape Research (WSL) for facilitating this research project in the Erlenbach catchment and for sharing hydro-climatic data. We are also particularly grateful to Ilja van Meerveld for helpful discussions. JLAK was funded through an ETH Zurich Postdoctoral Fellowship.





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





**Table 1: We assessed 15 environmental controls grouped into five different categories: seasonality indicators, relative input concentrations, antecedent wetness conditions, event characteristics, and event-water contributions. Minimum and maximum values indicate the ranges observed in the dataset. Relative input concentrations are specific for every solute (and not available for all solutes and events), whereas all other controls do not differ between individual solutes. Event-water contributions were only assessed for 22 out of 30 events. Groundwater levels are expressed as negative values so that the maximum corresponds to the wettest conditions, consistent with the other wetness indicators.**

| Variable | Description | Min. value | Max. value |
|---|---|---|---|
| Seasonality indicators: | | *cold ↔ warm* | |
| $T_{event}$ | Average air temperature during the event (°C) | -0.31 | 15.67 |
| $T_{24h}$ | Average air temperature in the 24h before the event (°C) | 6.23 | 18.45 |
| $SW$ | Proximity to midyear (-) | 0.31 | 1.00 |
| Relative input concentrations: | | *low ↔ high* | |
| $c_P/c_{Q_0}$ | Ratio between solute concentrations in precipitation and baseflow (solute-specific) (-) | 8.6e-4 | 3.91 |
| Antecedent wetness conditions: | | *dry ↔ wet* | |
| $GW_{ini}$ | Initial groundwater level (cm) | -76.50 | -9.70 |
| $AP_{7d}$ | Amount of precipitation in the 7 days before the event (mm) | 4.80 | 112.40 |
| $Q_{ini}$ | Baseflow before the onset of the event (mm h$^{-1}$) | 7.7e-04 | 0.11 |
| Event characteristics: | | *small ↔ large* | |
| $P_{intensity}$ | Maximum precipitation within 4h (mm) | 7.10 | 27.70 |
| $Q_{tot}$ | Total amount of event discharge (mm) | 0.82 | 28.48 |
| $P_{tot}$ | Total amount of precipitation (mm) | 8.00 | 67.70 |
| $Q_{tot}/P_{tot}$ | Runoff coefficient | 0.06 | 0.47 |
| $\Delta Q$ | Max. discharge change (L s$^{-1}$) | 24.82 | 901.45 |
| $\Delta t_{event}$ | Event duration (days) | 0.52 | 4.69 |
| Event-water contributions: | | *pre-event water ↔ event-water* | |
| $Q_e/P$ | Event water in streamflow as fractions of precipitation (event runoff coefficient) (-) | 4.9e-03 | 0.17 |
| $Q_e/Q$ | Event water in streamflow as fractions of discharge (-) | 0.04 | 0.46 |



**Table 2: Median solute concentrations and their upper/lower quartiles in streamwater, groundwater, and precipitation in the Erlenbach catchment and in soil water in the adjacent Studibach catchment. Groundwater solute data were collected at one to three different sampling times at two pumping wells located in the upper part of the catchment. These are probably not representative of groundwater concentrations throughout the catchment, but still provide a rough indication of which solutes dominate groundwater. Concentrations in precipitation and streamwater were obtained from the time series recorded at the Erlenbach outlet, excluding months with snow. *N* indicates the number of samples the calculations are based on. EC was not analyzed in precipitation or soil water. Concentrations greater than 10 µg L$^{-1}$ are rounded to the nearest integer. Flux index calculations (see Eq. (1)) of the snow-free season are based on the same period as the analyzed time series and consequently do not represent long-term fluxes. Positive flux indices quantify the fraction of the output flux that was generated in the catchment, while negative flux indices quantify the fraction of the input flux retained in the catchment. A value of 0 is obtained if input and output fluxes balance. The ratio of precipitation to streamflow water fluxes during the snow-free period was 1.75 (compared to a ratio of 1.41 for the time from January 1$^{st}$ 2017 to December 31$^{st}$ 2018).**

| Solute | Streamwater concentration [solutes: µg L$^{-1}$, EC: µS cm$^{-1}$] (*N*=1916-4930, $N_{EC}$ = 99576) | Groundwater concentration [solutes: µg L$^{-1}$, EC: µS cm$^{-1}$] (*N*=2-6) | Precipitation concentration [µg L$^{-1}$] (*N*=817-975) | Flux indices of the snow-free period [-] | Soil water concentration at Studibach [µg L$^{-1}$] (*N*=41-102) |
|---|---|---|---|---|---|
| EC | 264 [222-292] | 389 [354-398] | n.a. | n.a. | n.a. |
| Ca | 47891 [40528-52835] | 50057 [46648-53880] | 1366 [948-1896] | 0.93 | 13553 [3461-31007] |
| Mg | 3076 [2494-3498] | 1767 [1503-2013] | 79 [59-112] | 0.93 | 13588 [3885-20429] |
| Na | 2338 [1711-2924] | 1590 [1220-1936] | 126 [91-226] | 0.74 | 781 [526-999] |
| Sr | 299 [230-347] | 556 [551-560] | 4.25 [2.9-6.4] | 0.95 | 99 [0.77-297] |
| Ba | 46 [34-55] | 61 [57-65] | 0.91 [0.65-1.47] | 0.91 | 36340 [9675-61548] |
| B | 8.5 [6.7-11] | 21[6.4-39] | 0.60 [0.31-1.22] | 0.70 | 12 [8.3-20] |
| SO$_4$ | 9157 [6542-14102] | 4944 [3557-6531] | 172 [64-472] | 0.92 | 892 [428-1865] |
| K | 805 [659-907] | 1197 [1101-1210] | 81 [39-208] | 0.13 | 527 [273-879] |
| Cl | 264 [183-370] | 891 [768-961] | 33 [14-75] | 0.33 | 739 [447-1319] |
| NO$_3$ | 527 [384-758] | 1672 [514-1840] | 368 [158-812] | -0.52 | 84 [9.5-565] |
| Fe | 4.6 [2.3-15] | 1.21 [0.61-2.07] | 1.40 [0.87-2.46] | 0.87 | 18 [6.6-109] |
| Mn | 0.18 [0.11-0.54] | 0.42 [0.25-1.8] | 0.21 [0.09-0.61] | 0.54 | 12 [4.0 -64] |
| Cr | 0.05 [0.04-0.07] | 0.13 [0.13-0.14] | 0.01 [0.01-0.03] | 0.62 | 0.47 [0.02-1.04] |
| Cu | 1.22 [0.98-1.45] | 0.28 [0.24-0.56] | 0.21 [0.09-0.46] | 0.41 | 2.68 [1.84-4.83] |






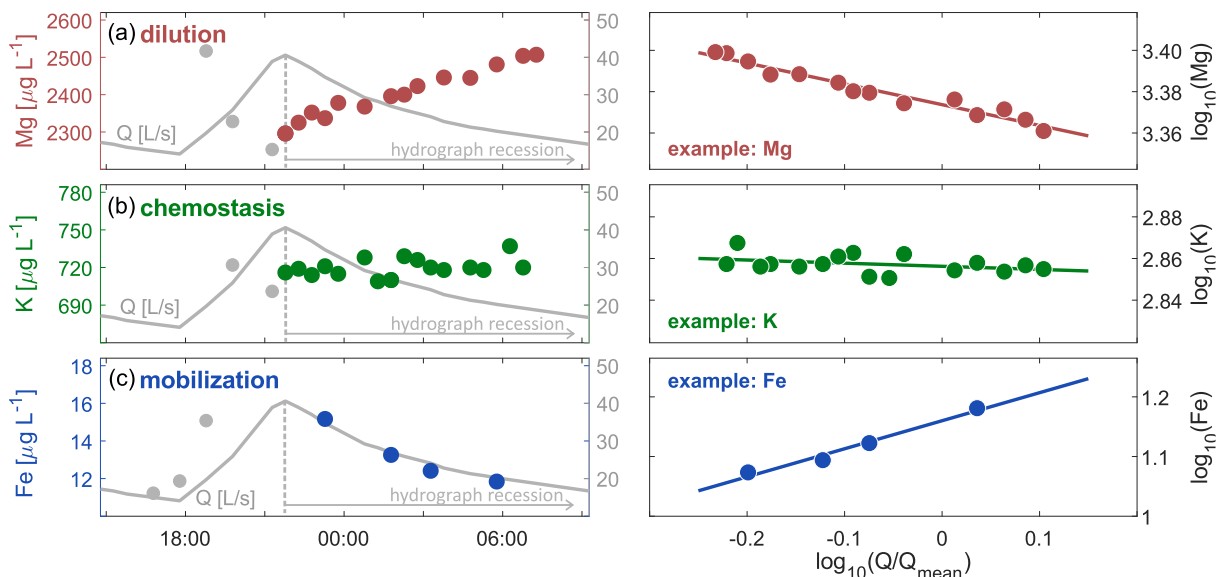

**Figure 1: Time series of discharge and solute concentrations (left panels), and corresponding cQ relationships (right panels) for a single recession, illustrating dilution behavior (Mg, top panels), chemostatic behavior (K, middle panels), and mobilization behavior (Fe, bottom panels). Solutes with negative cQ slopes (dilution, a) will have their lowest concentrations at high flows, and thus will**
**exhibit increasing concentrations during hydrograph recession. Because this concentration increase is usually less-than-proportional to the decrease in discharge, power-law cQ slopes are rarely steeper than -1. Solutes with cQ slopes near zero (chemostatic, b) do not vary systematically with discharge. Solutes with positive cQ slopes (mobilization, c) exhibit higher concentrations at high flows, and decreasing concentrations during hydrograph recession. Power-law cQ slopes steeper than 1 indicate that concentrations change more-than-proportionally to discharge.**




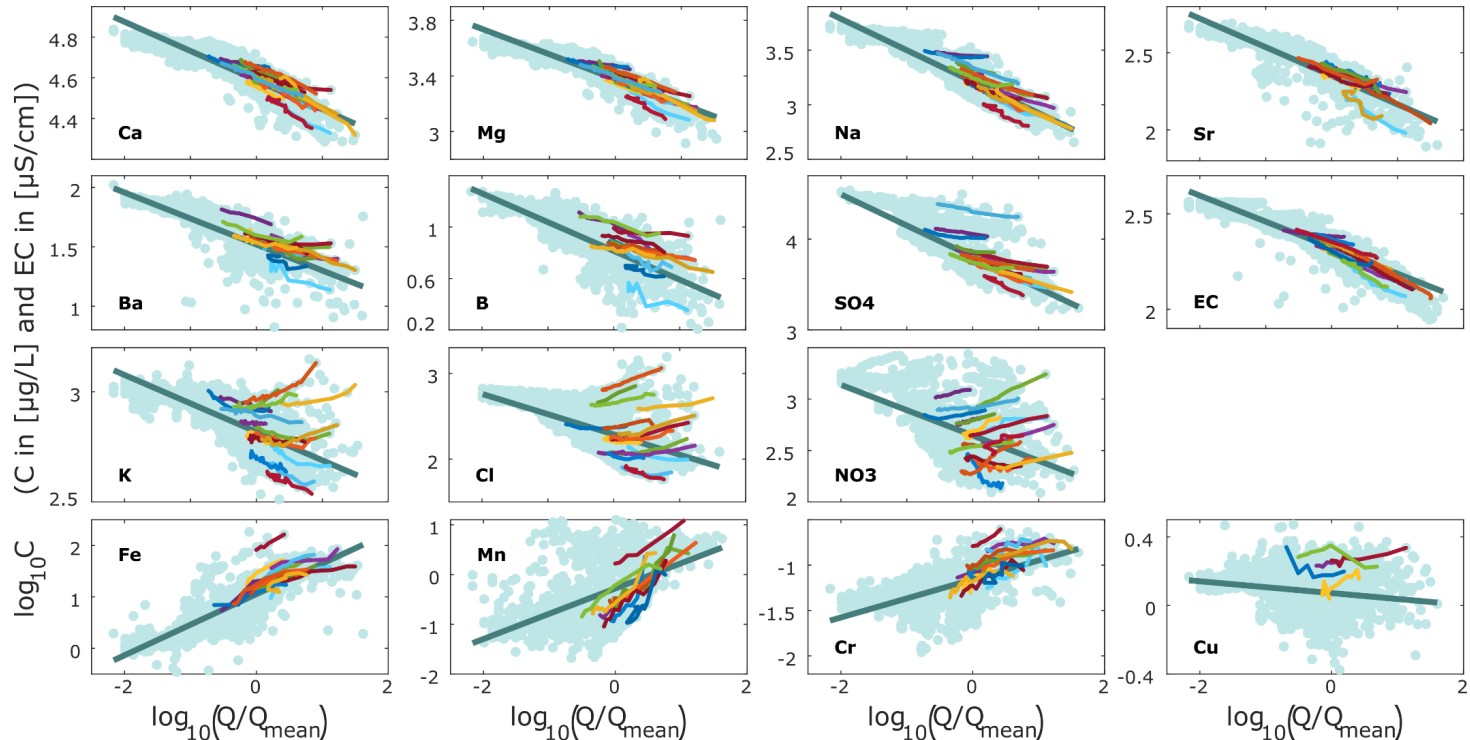

**Figure 2: Long-term cQ relationships from the entire recession time series (data points are shown in light blue and the dark blue line indicates the power-law fit), compared to the cQ behavior of individual events (colored lines; up to 20 events are shown for better visibility). Discharge is normalized by the average discharge of the time series. Most solutes exhibit a long-term dilution pattern, whereas the trace metals iron, manganese, and chromium exhibit a long-term mobilization pattern. The cQ relationships vary much more from event to event for some solutes (e.g. potassium, chloride, and nitrate) than for others (e.g. calcium, magnesium, sodium, and EC).**




**Figure 3: A comparison of event-scale and long-term cQ intercepts (a) and slopes (b). Blue circles represent intercepts and slopes for individual events, light-blue diamonds represent the averages of all events, and red bars indicate the slopes and intercepts of the long-term cQ relationships. If the red bar is close to the light-blue diamond, the long-term slope or intercept value is a good approximation of the average event slope or intercept. In panel (a), event cQ intercepts are expressed relative to the long-term average for better visual comparison of the solutes. Event-scale intercepts and slopes vary substantially for solutes in the lower half of the figure (from chloride to chromium), but vary little for most groundwater-sourced solutes. Because we normalized discharge by the mean discharge value, long-term intercepts approximate the average event-intercepts. Long-term slopes also approximate the event slopes for most groundwater-sourced solutes, but long-term slopes are more negative than event slopes for, e.g., chloride, nitrate, and manganese.**



**Figure 4: Scatter plots of event cQ slopes and intercepts of the 14 different solutes and EC (error bars indicate one standard error). Solutes from different dominant sources cluster and exhibit similar ranges of variability. (a) Groundwater-sourced solutes cluster closely around similar slopes and show little inter-event variability in both slopes and intercepts. (b) Intercepts and slopes of solutes with significant atmospheric input (i.e., chloride, nitrate, and potassium) vary substantially among events. (c) The slopes of trace metals are generally higher than those of the other solutes (indicating predominantly mobilization behavior), and are also highly variable among events. The uncertainty in the estimated slopes and intercepts is mostly smaller than the variability between events, indicating that the observed inter-event variability in slopes and intercepts reflects real-world behavior rather than sampling and measurement noise.**





**Figure 5: Weighted rank correlation coefficients expressing the dependence of event-scale cQ intercepts (a) and slopes (b) on different environmental controls (seasonality indicators, relative input concentrations, antecedent wetness conditions, event characteristics, importance of event-water contributions). Green fields indicate positive rank correlations, blue fields indicate negative correlations, and darker colors indicate stronger relationships. Only statistically significant ($p < 0.05$) correlation coefficients are displayed. Gray fields for EC indicate relationships that could not be assessed because EC was not measured in precipitation. For solutes featuring dilution patterns, e.g. calcium in panel (b), positive correlations indicate relationships that become more chemostatic when the controlling variable increases. For solutes with mobilization patterns, e.g. iron in panel (b), positive correlations indicate enhanced mobilization when the controlling variable increases. For meanings of abbreviations, please refer to Table 1.**





| dominant source | GROUNDWATER | PRECIPITATION | SOIL |
|---|---|---|---|
| cQ-behavior $\log_{10}(C)$ vs $\log_{10}(Q)$ | cations / anions | biological activity | |
| solute mobilization (cQ slope): | | | |
| long term | slope < 0 | slope < 0 | slope > 0 |
| event scale | slope < 0 | -1 < slope < 1 | slope > 0 |
| variability in cQ space: | | | |
| long term | well-confined | high scatter | high scatter |
| event scale | low variability | high variability | some variability |
| potential modulators of event-scale behavior | *ionic form:* lower variability for cations due to exchange buffering | *biological activity:* increases the variability | presence as *nanoparticulates:* likely affects the solute mobilization during events |
| drivers & controls of inter-event variability: *slope* | warm, dry, small events, more pre-event water / cold, wet, small events, more event water — seasonality indicators & event characteristics | dry, higher $c_p$, large events, more event water / wet, lower $c_p$, small events, more pre-event water — antecedent conditions & event-water contributions | cold, wet (opposed for Fe) / warm, dry (opposed for Fe) — seasonality indicators & antecedent conditions |
| drivers & controls of inter-event variability: *intercept* | warm, dry / cold, wet — seasonality indicators & antecedent conditions | (warm,) dry / (cold,) wet — (seasonality indicators &) antecedent conditions | warm, dry / cold, wet — seasonality indicators & antecedent conditions |
| solutes (at Erlenbach) | Ca, Mg, Na, Sr, Ba, B, SO$_4$, EC | Cl, NO$_3$ | Fe, Mn, Cr |
| mixed behavior | long-term cQ scatter and inter-event variability increases with contribution from precipitation, e.g. K. | | long-term behavior becomes more chemostatic with increasing contributions from precipitation, mobilization behavior on the event scale. |

**Figure 6: Schematic overview of cQ behavior based on dominant solute sources and environmental controls at the Erlenbach catchment. Solutes originating from similar sources generally behave similarly, and their cQ patterns are mostly sensitive to the same environmental controls. The light-blue patches in cQ behavior show the variability in the long-term cQ data, with average long-term cQ relationships indicated by the blue lines. Red lines show cQ relationships for individual events and indicate the degree of inter-event variability.**

695