# Peer review of "Concentration-discharge relationships vary among hydrological events, reflecting differences in event characteristics"

_Hydrology and Earth System Sciences, 2019_

## Referee Comment (RC1) · Anonymous Referee #1 · 31 Jan 2020

General comments:

This study examines the concentration-discharge (c-Q) relationships of several solutes at the scale of individual storm events and across the entire two-year study period in the Swiss Erlenbach catchment. The authors use the similarities and differences among solute c-Q slopes and intercepts to make inferences about the timing and hydrologic sources of streamwater in the catchment. The study also correlates the c-Q slope and intercept values with a wide variety of environmental controls to identify the most important regulators of solute transport within the Erlenbach catchment.

I commend the authors for undertaking a sampling regime intense enough in both its

frequency and duration to permit the development of such a unique dataset. High-frequency precipitation and stream sampling can be difficult during a single event, not to mention across multiple events. It is unfortunate that the nature of the Erlenbach catchment's hydrology was such that the study's sampling regime could not fully capture the c-Q responses across the complete hydrograph of the sampled storms. However, the authors made a convincing case for constraining their analysis to only the recession limb data, and they were mostly careful to not extent their inferences beyond where their limited dataset would allow them to go. I include the caveat "mostly" because I do question the description of a two-year, growing season only dataset as "long-term". This is perhaps a minor quibble, but I think it would be more accurate to describe the full dataset as "interannual" rather than "long-term". Overall, the authors present an interesting study of c-Q relationships at the catchment scale, highlighting important differences in biogeochemical responses observed across a range of temporal scales. This manuscript may be acceptable for publication in HESS, provided the authors address the specific comments outlined below.

Specific comments:

L99: How long did the precipitation funnel sit out in the open prior to the onset of rainfall? I am curious whether some of the solute concentrations in the early precipitation samples during events might have been biased by the addition of dry deposition on the funnel surface.

L121-122: Related to my previous comment, I'm curious how often it was the case that the first precipitation sample collected during an event was classified as an outlier. It might not affect your overall results much, but depending on how important a source dry deposition is in the study catchment, it might be worth considering.

Also, I think it's important to provide additional details about the identification of outliers in the dataset. Was there a threshold that you set, or did you truly just "eyeball" the dataset? For example, how did you identify outliers in the case of solutes that had more

variable (less tight) spreads? What proportion of the entire dataset was identified as outliers?

L125 and L134: The word "aggregated" is kind of vague. Does this mean you averaged the data? Calculated the median?

L174-175: Did you consider how this flux index might be affected by accumulated dry deposition on your precipitation collector? This would be more important for some ions (e.g., nitrate, chloride, maybe sulfate) compared to others.

L205-210: How many of the 30 events that you sampled fell into the category of being "not well constrained"? I can understand why you would want to limit your analysis to only those events for which the c-Q relationships were relatively straightforward, but this approach also kind of seems a little like "cherry picking" to me... From a practical standpoint, I completely understand the need to make such decisions about whether and when to exclude data (assuming they constitute a small percentage of the overall dataset) but I wonder if by limiting your dataset in this way, it also means that you're excluding some potentially important information about biogeochemical processes at event timescales. Those high RSEs are caused by something, and if they are attributable (even in part) to environmental and/or hydrologic drivers, there could be some very useful insights to be mined out of that variability.

L678-680: Somewhat related to the previous comment: do you seen smaller uncertainties within events relative to the variability between events because you have removed from your analysis the storms with elevated RSEs?

---

## Referee Comment (RC2) · Anonymous Referee #2 · 14 Feb 2020

General comments:

This is a well written, well presented, and interesting study on the cQ relationships of 14 different solutes in the Swiss Erlenbach catchment. The study sheds new light on the variability of these relationships among individual hydrological events. The resulting patterns and contrasts among the various solutes are clearly visualized. Perhaps the only shortcoming of this study is the lack of some more data/information like pH to better understand the role of biogeochemical processes in controlling the cQ relationships. This study calls for research to further link such patterns with detailed biogeochemical process studies and hypotheses.

[Figure]

Specific comments:

1. Introduction: already shed some light on the measurement approach (automated field lab) near the end of the introduction as it remains puzzling for the reader at the objectives how then such high resolution datasets were obtained for so many events.

2. Section 2.2. Dataset: I miss information on some essential parameters like silica, alkalinity, and pH:

a. Was silica measured by ICP-MS but below detection? That seems surprising. Or was it not measured, or perhaps not feasible with ICP-MS? Because silica is usually a very useful element in cQ studies.

b. Why was Alkalinity not also measured? Or at least perhaps on weekly basis when the lab in the field was visited and samples collected? Some background information on Alkalinity is welcome to better understand the biogeochemistry in this catchment. Only towards the end of the paper is it suggested that bicarbonate is the main (counter) anion (line 406).

c. Why was pH not measured with a sensor alike EC? Same request as for Alkalinity: some information is helpful for better understanding. Furthermore, variations in pH may be highly relevant to better understand variations in trace metals like Fe, Mn, Cr. This could also be added to the discussion/outlook. pH sensors I think should be sure be included in a downsized toolbox as discussed in section 5.

3. Section 3.1. Dataset: lines 240-245: what is meant with "sufficient .. measurements" and "were analysable"? Is it meant that samples with concentrations below the detection limit could not be used?

4. Section 3.2. Line 269: Clarify which solutes precisely are meant with "these solutes" and at line 266 with "weathering products". All the aforementioned solutes in this paragraph, or only Mg/Na/SO4?

5. General comment about Fe levels and its redox state (also in relation to lines 378-

379: nanoparticulates of trace metals). As pH is not provided it is difficult to better understand the Fe chemistry. Under acidic conditions the Fe could be leached from soil as Fe(III) and kept as ion in solution but at higher pH and also when Fe partially occurs as Fe(II) (after reduction in organic rich soils?) one would expect oxidation to Fe(III) in the stream and subsequent formation of (micro) iron flocs that might be removed by the automated field lab that filtrates the samples prior to ICP-MS analysis. Were also non-filtered samples analysed on Fe now and then to conclude anything on the potential role of particulate iron oxides in the stream? And how this may have affected cQ relationships?

6. Lines 346-348: what about the potential role of spatially and temporal varying evapotranspiration and also dry deposition?

7. Line 358: Clarify what it meant with Fe and Mn bound to organic material? Is meant that Fe and Mn as ions are bound (only?) to organic material? Can Mn and Fe not also or perhaps more importantly occur in the soils as Mn- and Fe-oxide minerals?

8. Lines 419-429: This part of the discussion on the behaviour of trace metals may profit from further discussion and information on the role of temperature (rate of microbial reactions; temperature dependent sorption of trace metals) and redox conditions (does wetting for example lead to suboxic conditions deeper in these soils?). Also information on the pH variations would be very welcome to better understand the metal behaviour.

9. Section 5: lines 475-481: especially in relation to understand metal behaviour employing also a pH sensor is essential.

---

## Author Comment (AC2) · 20 Feb 2020

**Response to the interactive comment of Anonymous Referee #2 on**

**"Concentration-discharge relationships vary among hydrological events, reflecting differences in event characteristics" by Julia L. A. Knapp et al.**

*General comments:*
*This is a well written, well presented, and interesting study on the cQ relationships of 14 different solutes in the Swiss Erlenbach catchment. The study sheds new light on the variability of these relationships among individual hydrological events. The resulting patterns and contrasts among the various solutes are clearly visualized. Perhaps the only shortcoming of this study is the lack of some more data/information like pH to better understand the role of biogeochemical processes in controlling the cQ relationships. This study calls for research to further link such patterns with detailed biogeochemical process studies and hypotheses.*

We thank Reviewer #2 for her/his positive comments on our manuscript. Please find our responses regarding additional data below.

*Specific comments:*
*1. Introduction: already shed some light on the measurement approach (automated field lab) near the end of the introduction as it remains puzzling for the reader at the objectives how then such high resolution datasets were obtained for so many events.*

Accepted. We will add a sentence at the end of the introduction mentioning the automated field laboratory: "In this study we analyzed high-frequency measurements of 14 different solutes ranging from major ions to trace metals that we obtained from an automated field laboratory set up at the outlet of a pre-Alpine catchment (von Freyberg et al., 2017). We quantified multi-year cQ relationships from the snow-free periods of a 2-year data set,…"

*2. Section 2.2. Dataset: I miss information on some essential parameters like silica, alkalinity, and pH:*
*a. Was silica measured by ICP-MS but below detection? That seems surprising. Or was it not measured, or perhaps not feasible with ICP-MS? Because silica is usually a very useful element in cQ studies.*

Acknowledged. We did not measure silica, because only a small part of it is available in dissolved form. The majority is present in streamwater as colloid polysilicic acid and is thus filtered out during the pre-processing of the sample.

*b. Why was Alkalinity not also measured? Or at least perhaps on weekly basis when the lab in the field was visited and samples collected? Some background information on Alkalinity is welcome to better understand the biogeochemistry in this catchment. Only towards the end of the paper is it suggested that bicarbonate is the main (counter)anion (line 406).*

Alkalinity was not measured at high frequency because this measurement could not be automated and thus not measured in our field laboratory set-up. Within the "National

Long-term Surveillance of Swiss Rivers" (NADUF), a composite streamwater sample is collected weekly and analyzed for major ions at the Erlenbach outlet. Based on NADUF data from 2017 and 2018, we find that alkalinity is strongly correlated with calcium and magnesium concentrations at Erlenbach (see figure below), and thus relationships between alkalinity and discharge would likely be very similar to those observed for calcium and magnesium. We agree that some information on the expected chemistry at the site would be helpful. We will include the following text in the site description: "The underlying geologic formation is Flysch, and the highly layered bedrock consists of limestone, claystone, marl, and shale, as well as conglomerate and calcareous sandstone (Zobrist, 2010). Groundwater chemistry is thus dominated by calcium, magnesium, and their counter-anion, bicarbonate."

[Figure]

c. Why was pH not measured with a sensor alike EC? Same request as for Alkalinity: some information is helpful for better understanding. Furthermore, variations in pH may be highly relevant to better understand variations in trace metals like Fe, Mn, Cr. This could also be added to the discussion/outlook. pH sensors I think should be sure be included in a downsized toolbox as discussed in section 5.

pH was measured at the same frequency as EC in streamwater. However, we experienced problems with the pH sensor after May 2018, so we refrain from using pH measurements in our analysis. At Erlenbach, streamwater pH is buffered due to the high

concentrations of bicarbonate and varies between 7.6 and 8.3. Short-term drops in pH can be observed during precipitation events, with quick recovery to the initial pH values as soon as rainfall stops.

We had initially analyzed pH, but found no behavior comparable with any of the trace metals (i.e., no similar sensitivities of pH and any trace metals to environmental controls). The cQ slopes and intercepts of the trace metals were also not related to any metric calculated from pH (relative or absolute change in pH during the event, cQ slope of pH etc. tested as environmental control). Instead, those metrics seemed to be more closely related to the patterns of the groundwater solutes (e.g., Ca, Mg, Na). We believe this is because the mobilization of trace metals is related to soil pH at the relevant location, but this can differ substantially from the pH variations observed in streamwater. Also, streamwater pH is probably buffered substantially by bicarbonate, which is groundwater-derived. Consequently, variations in streamwater pH do not seem related to the mobilization of trace metals. However, we acknowledge that this may be different in other, less buffered systems. We will consequently add the following statement to the outlook in Section 5 (lines 475-481, please also see comment #9): "… suitable proxy for iron and some other trace metals. pH measurements may also provide helpful insights into the mobilization of trace metals from soil layers."

*3.Section 3.1. Dataset:*
*lines 240-245: what is meant with "sufficient .. measurements" and "were analysable"? Is it meant that samples with concentrations below the detection limit could not be used?*

No, but because ICP-MS analysis was only performed on every second sample, fewer concentration data were available for these solutes. If any of these few data points had to be excluded because it was identified as an outlier, this resulted in a number of event recessions with fewer than the required 5 measurements for these solutes. Furthermore, specific solutes and events were excluded, because the relative standard errors of the cQ slopes or intercepts were too high. We will clarify this in the revised version of the manuscript by changing the sentence to: "While IC measurements were available for all 30 hydrologic events, not all events had the required 5 sample points for all of the cations analyzed by ICP-MS (i.e., boron, barium, iron, manganese, chromium, strontium, and copper). Furthermore, some events had to be excluded for individual solutes due to high relative standard errors of cQ slopes and/or intercepts (see Section 2.5). This resulted in 24 to 30 usable events for major ions measured with the IC. More than 20 events were evaluated for all other solutes, except manganese and copper, for which only data from 17 and 11 events were usable, respectively."

*4. Section 3.2. Line 269: Clarify which solutes precisely are meant with "these so-lutes" and at line 266 with "weathering products". All the aforementioned solutes in this paragraph, or only Mg/Na/SO4?*

The statement regarding a heterogeneous distribution of groundwater solutes was intended to include all groundwater solutes, not just Mg/Na/SO4. We will make this clearer by amending the text as follows (lines 265-269): "In general, groundwater concentrations of weathering products are likely to be highly heterogeneous in the Erlenbach catchment due to spatially variable contributions from the geochemically complex Flysch bedrock (Fischer et al., 2015; Kiewiet et al., 2019). Precipitation concentrations of  weathering products are low,…"

*5. General comment about Fe levels and its redox state (also in relation to lines 378-379: nanoparticulates of trace metals). As pH is not provided it is difficult to better understand the Fe chemistry. Under acidic conditions the Fe could be leached from soil as Fe(III) and kept as ion in solution but at higher pH and also when Fe partially occurs as Fe(II) (after reduction in organic rich soils?) one would expect oxidation to Fe(III) in the stream and subsequent formation of (micro) iron flocs that might be removed by the automated field lab that filtrates the samples prior to ICP-MS analysis. Were also non-filtered samples analysed on Fe now and then to conclude anything on the potential role of particulate iron oxides in the stream? And how this may have affected cQ relationships?*

pH at the Erlenbach was always above 7.6 and ranged up to 8.2. As pointed out by the reviewer, the presence of (micro) iron flocs in streamwater is likely, and these may have been removed during filtration. Unfortunately, unfiltered water samples have not been analyzed, because filtration is an automatic step in our setup, and because the ICP-MS is easily damaged when analyzing unfiltered streamwater samples, as these are usually highly turbid particularly at times of interest (i.e., when flow is high).

*6. Lines 346-348: what about the potential role of spatially and temporal varying evapotranspiration and also dry deposition?*

Agreed. We will add an explanation to the manuscript: "… may result from varying degrees of atmospheric deposition, evapotranspiration, and dry deposition, resulting in temporally and spatially variable concentrations of chloride,…."

*7. Line 358: Clarify what it meant with Fe and Mn bound to organic material? Is meant that Fe and Mn as ions are bound (only?) to organic material? Can Mn and Fe not also or perhaps more importantly occur in the soils as Mn- and Fe-oxide minerals?*

Agreed. We will re-formulate this as "… where they are bound to organic material or present as oxides."

*8. Lines 419-429: This part of the discussion on the behaviour of trace metals may profit from further discussion and information on the role of temperature (rate of microbial reactions; temperature dependent sorption of trace metals) and redox conditions (does wetting for example lead to suboxic conditions deeper in these soils?). Also information on the pH variations would be very welcome to better understand the metal behaviour.*

Acknowledged. We agree that the factors mentioned by the reviewer are important for the mobilization of metals from soils. Unfortunately, we have no spatial information on soil temperature, redox conditions, or soil pH. We will, however, include a note on this at the end of the paragraph as suggested: "As metal complexation and mobilization is known to depend on various factors such as pH and redox conditions in the soil layer (Gotoh and Patrick, 1972, 1974), further field measurements are necessary to better understand the mobilization of trace metals from soil layers during hydrologic events."

*9. Section 5: lines 475-481: especially in relation to understand metal behaviour employing also a pH sensor is essential.*

Agreed. We will add the following in the revised manuscript: "… The analysis of trace metals is not available through automated sensors, but in many cases DOC can be a suitable proxy for iron and some other trace metals. pH measurements may also provide helpful insights into the mobilization of trace metals from soil layers." (Please also see our response to comment #2c)

**References:**

von Freyberg, J., Studer,B., and Kirchner, J. W.: A lab in the field: high-frequency analysis of water quality and stable isotopes in stream water and precipitation, Hydrol. Earth Syst. Sci., 21, 1721-1739, https://doi.org/10.5194/hess-21-1721-2017, 2017.

Gotoh, S., and Patrick, W. H.: Transformation of Manganese in a Waterlogged Soil as Affected by Redox Potential and pH. *Soil Science Society of America Journal*, *36*(5), 738-742, 1972.

Gotoh, S., and Patrick, W. H.: Transformation of iron in a waterlogged soil as influenced by redox potential and pH. *Soil Science Society of America Journal*, *38*(1), 66-71, 1974.

---

## Author Response (AR2)

Response to Reviewers – Point-by-point response to reviewer comments on "Concentration-discharge relationships vary among hydrological events, reflecting differences in event characteristics" by Julia L. A. Knapp et al.

We would like to thank two anonymous reviewers for helpful comments on our manuscript. The point-by-point reply to the comments is given below. The comments provided by the reviewers are shown in italics, and our responses in regular font in blue. Changes made to the manuscript are indicated by underlined text. Line numbers in the responses refer to the updated manuscript with tracked changes marked.

Contrary to the reply we posted to the comments from reviewer #1, we will use the term "two-year" to describe the results from the full dataset, to follow the recommendations by the editor.

Please note that we have also had to update our dataset due to a registration error in the discharge timeseries, which resulted in a shift of one hour for all data points. This does not affect the interpretation of our results, but changes all numbers by the second or first decimal. We have therefore updated all figures and tables in the manuscript.

**1) Response to the interactive comment of Reviewer #1**

**General comments:**

This study examines the concentration-discharge (c-Q) relationships of several solutes at the scale of individual storm events and across the entire two-year study period in the Swiss Erlenbach catchment. The authors use the similarities and differences among solute c-Q slopes and intercepts to make inferences about the timing and hydrologic sources of streamwater in the catchment. The study also correlates the c-Q slope and intercept values with a wide variety of environmental controls to identify the most important regulators of solute transport within the Erlenbach catchment. I commend the authors for undertaking a sampling regime intense enough in both its frequency and duration to permit the development of such a unique dataset. High-frequency precipitation and stream sampling can be difficult during a single event, not to mention across multiple events. It is unfortunate that the nature of the Erlenbach catchment's hydrology was such that the study's sampling regime could not fully capture the c-Q responses across the complete hydrograph of the sampled storms. However, the authors made a convincing case for constraining their analysis to only the recession limb data, and they were mostly careful to not extent their inferences beyond where their limited dataset would allow them to go. I include the caveat "mostly" because I do question the description of a two-year, growing season only dataset as "long-term". This is perhaps a minor guibble, but I think it would be more accurate to describe the full dataset as "interannual" rather than "long-term". Overall, the authors present an interesting study of c-Q relationships at the catchment scale, highlighting important differences in biogeochemical responses observed across a range of temporal scales. This manuscript may be acceptable for publication in HESS, provided the authors address the specific comments outlined below.

We thank the reviewer for her/his positive assessment of our study. We understand the concerns regarding the term "long-term" for a two-year data set. The terminology was intended to more clearly separate observations based on the longer dataset from those

of individual hydrologic events. Nevertheless, we understand that it may be misleading. Following the recommendation from the editor, we use the term "two-year" instead in the revised version.

**Specific comments:**

L99: How long did the precipitation funnel sit out in the open prior to the onset of rainfall? I am curious whether some of the solute concentrations in the early precipitation samples during events might have been biased by the addition of dry deposition on the funnel surface.

The collection and analysis of precipitation was automated, like the streamwater sampling. Therefore, the funnel was not cleaned and rinsed before every sample. However, rainfall events at Erlenbach occur roughly every second day, and the effect of dry deposition accumulating on the funnel was likely small. To account for the effect of dry deposition, we used (rainfall-) volume-weighted precipitation concentrations in our calculations (Tables 1 and 2). Small-volume samples at the start of a rain event with high solute concentrations due to dry deposition therefore had little weight in the calculation. Furthermore, our analysis is mostly based on the streamwater samples. The precipitation samplings procedure is described for completeness, and because the solute concentrations in precipitation are investigated as potential environmental drivers to explain the observed cQ slopes and intercepts (sections 2.6 and 3.4).

L121-122: Related to my previous comment, I'm curious how often it was the case that the first precipitation sample collected during an event was classified as an outlier. It might not affect your overall results much, but depending on how important a source dry deposition is in the study catchment, it might be worth considering. Also, I think it's important to provide additional details about the identification of outliers in the dataset. Was there a threshold that you set, or did you truly just "eyeball" the dataset? For example, how did you identify outliers in the case of solutes that had more variable (less tight) spreads? What proportion of the entire dataset was identified as outliers?

We excluded very few precipitation samples from our analysis, usually only if there was a known problem with the precipitation collector. Precipitation solute concentrations were generally very variable, even within an individual rain event. Generally, we observed a gradual decrease in solute concentrations as the rain event progressed. A possible explanation of this gradual decrease in precipitation solute concentrations could be due to the wash-off of dry deposition from leaves and the rain-out of aerosols in the atmosphere, and therefore likely reflect the normal processes occurring in the landscape.

Eliminating outliers in streamwater samples was mainly based on visual inspection. During the hydrograph recession, solute concentrations at Erlenbach generally change gradually from sample to sample (given our very high sampling frequency). Sudden jumps and outliers due to anomalous instrument behavior or sample handling were therefore often easy to detect. We furthermore compared among solute groups and instruments. Suspected anomalies in cation and anion data were compared across instruments, and if these anomalies were inconsistent across instruments, the respective data points were removed. We were generally very conservative in the removal of outliers.

**L125 and L134: The word "aggregated" is kind of vague. Does this mean you averaged the data? Calculated the median?**

Because streamwater sampling is instantaneous (not composite) in our field laboratory, we picked the closest 10-min data point for any quantity related to streamflow (discharge, EC in streamwater), air temperature and groundwater level. For the precipitation samples, the 10-min precipitation rates were cumulated from the start of the event or the last sampling time until the time of sampling. We clarify this in the revised manuscript: "For the purpose of this study, only every second EC measurement was used to match the 10-min measurement frequency of river discharge" (lines 128-129), and "For this purpose, we extracted those data that were closest to the sampling times of each streamwater sample from the 10-minute discharge, air temperature, and groundwater level time series. For the precipitation samples, associated precipitation amounts were calculated as cumulative sums from the 10-minute tipping bucket recordings." (lines 138-140)

**L174-175: Did you consider how this flux index might be affected by accumulated dry deposition on your precipitation collector? This would be more important for some ions (e.g., nitrate, chloride, maybe sulfate) compared to others.**

Acknowledged. It is indeed possible that dry deposition affected the flux budgets. However, dry deposition of soluble compounds \*should\* be part of the flux budgets, since they are real input fluxes that need to be taken into account if one wants an accurate picture of whether the catchment is a net source or sink of the solutes in question. In any case the effect should be small; given that it rains frequently at Erlenbach, the amount of dry deposition was probably relatively negligible (see explanation above).

L205-210: How many of the 30 events that you sampled fell into the category of being "not well constrained"? I can understand why you would want to limit your analysis to only those events for which the c-Q relationships were relatively straightforward, but this approach also kind of seems a little like "cherry picking" to me... From a practical standpoint, I completely understand the need to make such decisions about whether and when to exclude data (assuming they constitute a small percentage of the overall dataset) but I wonder if by limiting your dataset in this way, it also means that you're excluding some potentially important information about biogeochemical processes at event timescales. Those high RSEs are caused by something, and if they are attributable (even in part) to environmental and/or hydrologic drivers, there could be some very useful insights to be mined out of that variability.

Acknowledged. We excluded cQ relationships from events and solutes with high relative standard error (RSE) because they did not allow us to fit a power-law relationship, and thus would not allow for a consistent interpretation of the recession behavior. Nevertheless, we understand the concern that we may be excluding potentially important information. We thus redid the analysis with all calculated slope and intercept values (weighted by the inverse of the respective RSE) and we obtained a very similar result of the correlation analysis shown in Figure 5 of the manuscript.

The number of events excluded due to high RSEs was small for most solutes (usually none, and a maximum of 7 out of the 30 events), except manganese and copper (for which 13 and 17 events out of the 30 events had to be excluded because of high RSE, respectively). The generally low concentrations of these two solutes had high fractional variability, resulting in no clear power-law relationship for many events. We include a note on this in the revised version of the manuscript: "More than 20 events were evaluated for all other solutes, except manganese and copper. The concentrations of these two solutes were low and variable, resulting in no clear power-law relationship with discharge for many events. Consequently, only data from 17 and 11 events were usable for manganese and copper, respectively." (lines 251-254)

L678-680 (Figure 4): Somewhat related to the previous comment: do you seen smaller uncertainties within events relative to the variability between events because you have removed from your analysis the storms with elevated RSEs?

This is a valid concern. We have added a copy of Figure 4 (see figure below) including the uncertainties in slope and intercept of all events in the supplemental material, even those excluded from further analysis. We provide a note on this in the caption of Figure 4 in the main text. As you can see, this mostly does not invalidate our statement that "the uncertainty in the estimated slopes and intercepts is mostly smaller than the variability between events, …". An exception may be manganese.

Figure 4 updated. Scatter plots of event cQ slopes and intercepts of the 14 different solutes and EC (error bars indicate one standard error). Solutes from different dominant sources cluster and exhibit similar ranges of variability. (a) Groundwater-sourced solutes cluster closely around similar slopes and show little inter-event variability in both slopes and intercepts. (b) Intercepts and slopes of solutes with significant atmospheric input (i.e., chloride, nitrate, and potassium) vary substantially among events. (c) The slopes of trace metals are generally higher than those of the other solutes (indicating predominantly mobilization behavior), and are also highly variable among events. The uncertainty in the estimated slopes and intercepts is mostly smaller than the variability between events, indicating that the observed inter-event variability in slopes and intercepts reflects real-world behavior rather than sampling and measurement noise.

**2) Response to the interactive comment of Reviewer #2**

**General comments:**

This is a well written, well presented, and interesting study on the cQ relationships of 14 different solutes in the Swiss Erlenbach catchment. The study sheds new light on the variability of these relationships among individual hydrological events. The resulting patterns and contrasts among the various solutes are clearly visualized. Perhaps the only shortcoming of this study is the lack of some more data/information like pH to better understand the role of biogeochemical processes in controlling the cQ relationships. This study calls for research to further link such patterns with detailed biogeochemical process studies and hypotheses.

We thank Reviewer #2 for her/his positive comments on our manuscript. Please find our responses regarding additional data below.

**Specific comments:**

1. Introduction: already shed some light on the measurement approach (automated field lab) near the end of the introduction as it remains puzzling for the reader at the objectives how then such high resolution datasets were obtained for so many events.

Accepted. We will add a sentence at the end of the introduction mentioning the automated field laboratory: "In this study we analyzed high-frequency measurements of 14 different solutes ranging from major ions to trace metals that we obtained from an automated field laboratory set up at the outlet of a pre-Alpine catchment (von Freyberg et al., 2017). We quantified multi-year cQ relationships from the snow-free periods of a 2-year data set,..." (lines 73-75)

2. Section 2.2. Dataset: I miss information on some essential parameters like silica, alkalinity, and pH:

a. Was silica measured by ICP-MS but below detection? That seems surprising. Or was it not measured, or perhaps not feasible with ICP-MS? Because silica is usually a very useful element in cQ studies.

Acknowledged. We did not measure silica, because only a small part of it is available in dissolved form. The majority is present in streamwater as colloid polysilicic acid and is thus filtered out during the pre-processing of the sample.

b. Why was Alkalinity not also measured? Or at least perhaps on weekly basis when the lab in the field was visited and samples collected? Some background information on Alkalinity is welcome to better understand the biogeochemistry in this catchment. Only towards the end of the paper is it suggested that bicarbonate is the main (counter)anion (line 406).

Alkalinity was not measured at high frequency because this measurement could not be automated and thus not measured in our field laboratory set-up. Within the "National Long-term Surveillance of Swiss Rivers" (NADUF), a composite streamwater sample is collected weekly and analyzed for major ions at the Erlenbach outlet. Based on NADUF data from 2017 and 2018, we find that alkalinity is strongly correlated with calcium and magnesium concentrations at Erlenbach (see figure below), and thus relationships between alkalinity and discharge would likely be very similar to those observed for calcium and magnesium. We agree that some information on the expected chemistry at the site would be helpful. We will include the following text in the site description: "The underlying geologic formation is Flysch, and the highly layered bedrock consists of limestone, claystone, marl, and shale, as well as conglomerate and calcareous sandstone... Groundwater chemistry is thus dominated by calcium, magnesium, and their counter-anion, bicarbonate." (lines 84-85)